# Multi-Dimensional Collaborative Operation Model and Evaluation of Cascade Reservoirs in the Middle Reaches of the Yellow River

Xinjie Li [1,2,3], Qiang Wang [1,2,*], Yuanjian Wang [1,2,*], Hongtao Zhang [4], Jieyu Li [1,2] and Donglin Li [1,2]

1   Yellow River Water Conservancy Research Institute, Yellow River Water Conservancy Commission, Zhengzhou 450003, China; xin_wd@163.com (X.L.); hzlijy@163.com (J.L.); lidonglinl@163.com (D.L.)
2   Key Laboratory of Lower Yellow River Channel and Estuary Regulation, Ministry of Water Resources, Zhengzhou 450003, China
3   Engineering Department, Huanghe Science and Techology University, Zhengzhou 450063, China
4   School of Electrical Engineering, North China University of Water Resources and Hydropower, Zhengzhou 450003, China; zht1977@ncwu.edu.cn
*   Correspondence: hkywangqiang@163.com (Q.W.); wangyuanjian_yrcc@aliyun.com (Y.W.)

**Abstract:** Reservoir operation optimization is a technical measure for flood control and is beneficial owing to its reasonable and reliable control and application of existing water conservancy and hydropower hubs, while ensuring dam safety and flood control, as well as the normal operation of power supply and water supply. Considering the beneficial functions of reservoirs, namely flood control and ecological protection, this paper firstly established a two-objective optimal operation model for the reservoir group in the middle reaches of the Yellow River. We aim to maximize the average output of the cascade reservoir group and minimize the average change in ecological flow during the operation period under efficient sediment transport conditions, with the coordination degree of water and sediment as the constraints of reservoir discharge flows. The paper aims to construct an evaluation index system for reservoir operation schemes, apply a combined approach of objective and subjective evaluations, and introduce the gray target and cumulative prospect theories. By uniformly quantifying the established scheme evaluation index system, screening the reservoir operation schemes with the fuzzy evaluation method, and selecting the recommended scheme for each typical year, this paper provides a new scientific formulation of the operation schemes of reservoirs in the middle reaches of the Yellow River. The selected schemes are compared with actual data, demonstrating the effectiveness of joint reservoir operation and for multidimensional benefits in terms of power generation, ecology, and flood control.

**Keywords:** Yellow River basin; cascade reservoir; operation optimization; sediment transport; water and sediment coordination degree

## 1. Introduction

Reservoirs mainly undertake a variety of important tasks such as flood control, power generation, and water supply, but there is a synergistic competition relationship between different objectives. The means to scientifically and reasonably allocate water resources and achieve the maximum total benefits of reservoirs is the goal pursued by managers [1–6]. In sediment-rich rivers, reservoirs are mainly constructed to intercept large amounts of sediment upstream to ensure the normal function of downstream rivers; however, this leads to the siltation of reservoir capacity and seriously affects the service life of reservoirs [7]. In the past half century, due to the influence of human activities, the water and sediment characteristics of rivers have changed significantly, the regulation, storage capacity, and ecological environment of basins have transformed, and reservoir operation has affected the ecological environment [8–10]. Reservoir operation tends to be basin-based and multi-objective [11]. In the process of reservoir operation, considering not only the hydraulic

connection of upstream and downstream reservoirs but also the maximization of reservoir functions and benefits is necessary. Additionally, the maintenance of long-term effective reservoir capacity and the improvement of sedimentation patterns, including the impact of discharge flow on downstream flood control, water supply, and ecology, and the influence of outgoing sediment on the downstream river regime and riverbed must be considered. To this end, the future regulation of water and sediment in the basin for long-term reservoir service must focus on meeting the requirements for a balance between the water and sediment in the basin and give play to the comprehensive benefits of reservoirs without affecting the water–sediment–electricity multi-objective joint operation optimization.

Research on the joint water–sediment operation of reservoirs has played an important role in the construction of water conservancy projects and the optimal spatiotemporal allocation of resources in sediment-rich rivers. This is a typical multi-objective problem of a large system. The relevant research has developed greatly since it was proposed in the late 1980s, and it focuses on the selection of joint water–sediment operation objectives, as well as the construction, processing, and solving of multi-objective models. Since 1974, when the Sanmenxia reservoir adopted an operation mode of "storing clean water and discharging muddy flows, regulating water and sediment", scholars in China have started research on the joint water–sediment operation of reservoirs on sediment-rich rivers. In 1992, to minimize sedimentation in the downstream river, Du et al. embedded a sediment erosion and deposition computation model into the constructed stochastic dynamic programming model of joint water–sediment operation in the Sanmenxia reservoir. They verified the rationality of the current operation mode centered on the issue of "dimensionality reduction". However, they did not consider the siltation site and the development level of reservoir sediment due to the development level of the sediment discipline [12]. Zhu and Qiu established a multi-objective optimization model of power generation and siltation reduction for cascade reservoirs in the upper reaches of the Yellow River and transformed the multi-objective problem into a single-objective one by the weighting method; however, the siltation reduction weighting factor was not set clearly [13]. With the rapid development of the sediment discipline within hydropower energy, the joint water–sediment operation of reservoirs has become increasingly complicated. Many scholars have carried out theoretical and practical research on the contradiction between long-term beneficial use, sediment discharge, and siltation reduction in reservoirs. These studies include models of hyperpycnal flow [14,15], optimal joint water–sediment operation [16], and erosion/deposition computation [17]. Based on the contradiction between water storage and the sediment discharge of reservoirs, a multi-objective decision-making model of joint water–sediment operation with reservoir flood control, power generation, and shipping operations as sub-modules was constructed. The joint water–sediment operation model of the Three Gorges Reservoir made multi-objective decisions on the water storage time and operation mode at the end of flood season possible [18]. Further studies have focused on maximizing sediment runoff and power generation [19] and intelligent algorithm-particle swarm optimization to obtain multiple feasible solutions, which received noteworthy attention from researchers on joint water–sediment operation [20]. Lian et al. also used the thinking method of multi-objective planning in combination with a genetic algorithm and the neural network approach for the joint water–sediment operation of reservoirs [21].

The multi-objective joint optimal water–sediment operation of reservoirs is a high-dimensional, multi-objective, nonlinear, and complex optimization problem with multiple constraints and variables [22]. Common solutions are mathematical programming methods and artificial intelligence optimization methods [23,24]. These include genetic algorithms [25–27], particle swarm optimization [28,29], artificial neural networks [30,31], ant colony optimization [32,33], simulated annealing [34,35], support vector machines [36], and fuzzy computation [37]. Early joint optimal water–sediment operation of reservoirs was usually solved by transforming the multi-objective problem into a single-objective one through weighted non-dimensionalization [38]. Further studies have used the analytical

hierarchy process to determine the weights of objectives [39] and adopted a neural network prediction method to compute the amount of reservoir sedimentation [40].

The purpose of reservoir optimization scheduling is to use optimization methods to formulate optimal scheduling methods for the inflow process and comprehensive utilization requirements of the reservoir, in order to achieve better benefits. The optimization strategy aims to maximize energy production rather than income, which is related to the energy market in different contexts [41]. At present, large-scale reservoir joint operation has become a key focus in the field of reservoir operation, with more and more constraints and objective functions. Therefore, the optimization of algorithms has become one of the bottlenecks that constrain joint operation. Reddy and Kumar, based on the framework of particle swarm optimization, designed a multi-objective parallel optimization to take irrigation and power generation into account [42], which was further improved by Brouwer and Groenwold and Zhou et al. [43,44]. Building upon the theory of Pareto optimization could achieve satisfactory results for decision making [45]. Qin et al. constructed a multi-objective optimal operation model by analyzing the restrictive and competitive relationships between power generation and flood control [46]. Wang Xuebin et al. proposed an improved fast non-inferiority ranking genetic algorithm (ICGC-NSGA-II) based on individual constraint and population constraint techniques. They established a multi-objective operation model for cascade reservoirs in the lower reaches of the Yellow River by considering water for ecological functions and comprehensive utilization needs in different periods in the lower reaches. ICGC-NSGA-II allowed them to explore the relationship between water supply benefits, power generation benefits, and ecological benefits of the reservoirs [47]. With the interdisciplinary research and application of different disciplines, the research results of underground gas storage can also provide a new perspective for reservoir scheduling [48].

In summary, the existing research has explored modeling and solution techniques for the optimal operation of reservoirs, but the research and application of algorithms for solving high-dimensional objective optimization problems for reservoir operation are relatively limited. With the significant reduction in incoming water and sediment and the development of cascade reservoirs in the basin, the water–sediment contradiction in improving the comprehensive benefits of cascade reservoirs and the operation for flood control and siltation reduction is becoming more prominent. This contradiction is especially apparent for cascade reservoirs in sediment-rich rivers. In-depth research on the multi-objective optimal operation of water, sediment, electricity, and ecology of cascade reservoirs under variations in water and sediment is necessary. This study aims to establish a two-objective optimal operation model for the reservoir group in the middle reaches of the Yellow River, in order to maximize the average output of the cascade reservoir group and minimize the average change in ecological flow during the period of operation. Herein, efficient sediment transport and the coordination degree of water and sediment serve as the constraints of reservoir discharge flow. The study also aims to construct an evaluation index system for reservoir operation schemes to provide theoretical support for the optimal operation of cascade reservoirs in the Yellow River basin.

## 2. Construction of a Two-Objective Optimal Operation Model for the Cascade Reservoir Group

Complex hydraulic connections among reservoir areas and between major parameters in the optimal operation model of a cascade reservoir group, as well as constraints in the time distribution of parameters for different stages, are considered in the development of a two-objective operation model. The joint operation of the Sanmenxia and the Xiaolangdi reservoirs serves as a backbone water conservancy hub, and the Wanjiazhai reservoir complements the Wanjiazhai, Sanmenxia, and Xiaolangdi cascade reservoirs built in the middle reaches of the Yellow River. Together, they can enhance the mutual cooperation and interconnection among reservoirs and strengthen the subsequent dynamics for the operation of the Xiaolangdi reservoir. Taking the Wanjiazhai, Sanmenxia, and Xiaolangdi reservoirs as research subjects, a two-objective optimal operation model of cascade reservoirs, which uses

months as a basic operation period and considers both economic and ecological benefits, was constructed to empower the high-quality development of the Yellow River basin.

*2.1. Objective Function*

The operation model uses the water level of each reservoir at the end of the flood season in each period (month) as the decision variable and constructs a two-objective optimal operation model for the cascade reservoir group. The overall objective $O_1$ is:

$$O_1 = F(M_{11}, M_{12}) \tag{1}$$

where $M_{11}$ is the economic benefit target, i.e., the maximum average output during the period of operation of the reservoir group. $M_{12}$ is the ecological benefit target, i.e., the minimum average change in ecological flow of each reservoir during the reservoir group's period of operation.

$$M_{11} = \max\left(\frac{1}{N}\sum_{n=1}^{N}\frac{1}{T}\sum_{t=1}^{T} A_n Q_{n}{}^{t}{}_{out}\Delta H_n{}^t\right) \tag{2}$$

$$A_n = 9.81\eta_n \tag{3}$$

where $N$ is the total number of reservoirs in the reservoir group; $T$ is the total number of periods during the period of operation; $t$ is the number of periods; $n$ is the reservoir number; $A_n$ is the comprehensive output coefficient of the hydropower station of the $n$th reservoir; $Q_n{}^t{}_{out}$ is the average outflow of the $n$th reservoir during the period (month) $t$ in $m^3/s$; $\Delta H_n{}^t$ is the average head of the $n$th reservoir during the period (month) $t$ in m; $\eta_n$ is the efficiency coefficient of the hydraulic turbine generator of the $n$th reservoir.

$$M_{12} = \min\left[\frac{1}{N}\sum_{n=1}^{N}\frac{1}{T}\sum_{t=1}^{T}\left(\frac{Q_n{}^t{}_{out} - Q_n{}^t{}_{AEF}}{Q_n{}^t{}_{AEF}}\right)^2\right] \tag{4}$$

where $t$ is the number of periods; $n$ is the reservoir number; $Q_n{}^t{}_{AEF}$ is the suitable ecological flow of the downstream river of the $n$th reservoir in the period (month) $t$ in $m^3/s$; $M_{12} \in [0, 1]$ is inversely proportional to the ecological benefit. The appropriate ecological flow of the downstream river in each period is obtained using the monthly frequency computation method.

*2.2. Constraints*

The constraints considered in the model include the water volume constraint of the cascade reservoirs, the water balance and level constraints, and the outflow and output constraints.

2.2.1. Water Volume Constraint of Cascade Reservoirs

A certain hydraulic connection between the cascade reservoirs is defined as where the outflow from the upstream reservoir and the inflow to the downstream reservoir satisfy the following constraints:

$$Q_{n+1}{}^t{}_{in}\Delta t = Q_n{}^t{}_{out}\Delta t + W_n^t \tag{5}$$

where $t$ is the number of periods; $n$ is the reservoir number; $Q_{n+1}{}^t{}_{in}$ is the average inflow of the $(n+1)$th reservoir during the period $t$ in $m^3/s$; $\Delta t$ is the time during the period $t$ in s; $W_n{}^t$ is the interval inflow between the $n$th and $(n+1)$th reservoirs during the period $t$ in $m^3$.

### 2.2.2. Water Balance Constraints

The conversion of periods is achieved through a water balance equation in which the reservoir capacity and inflow and outflow at the beginning and end of each period of a single reservoir satisfy the following constraints:

$$V_n^{t+1} - V_n^t = (Q_{nin}^t - Q_{nout}^t)\Delta t \tag{6}$$

where $t$ is the number of periods; $n$ is the reservoir number; $V_n^t$ is the initial reservoir capacity of the $n$th reservoir during the period $t$ in billion m$^3$; $V_n^{t+1}$ is the initial reservoir capacity (the final reservoir capacity during the period $t$) of the $n$th reservoir during the periods $t+1$ in billion m$^3$; $Q_{nin}^t$ is the average inflow of the $n$th reservoir during the period $t$ in m$^3$/s.

### 2.2.3. Water Level Constraint

Reservoirs at the beginning of the design from the dam safety perspective have a set normal storage level, high water level for flood control, flood-limited water level, and dead water level. Reservoirs in different stages of operation have different requirements for water level. Therefore, the water level amplitude is specified based on safety considerations. The water level during the period $t$ meets the following conditions:

$$Z_{n\min}^t \le Z_n^t \le Z_{n\max}^t \tag{7}$$

where $t$ is the number of periods; $n$ is the reservoir period; $Z_n^t$ is the initial water level of the $n$th reservoir during the period $t$ in m. $Z_{n\max}^t$ and $Z_{n\max}^t$ are the upper and lower water level limits of the $n$th reservoir during the period $t$, respectively, in m. In this paper, the water level constraint for each period is set according to the upper and lower water level limits for each period and the reservoir operation rules in actual reservoir operation.

### 2.2.4. Outflow Constraint

The outflow constraint conditions should consider the needs of power generation, flood control, ice-jam flood prevention, water supply, etc. The outflow at any moment during the period $t$ should meet the following conditions:

$$Q_n{}^t{}_{\min} \le Q_n{}^t{}_{out} \le Q_n{}^t{}_{\max} \tag{8}$$

where $t$ is the number of periods; $n$ is the reservoir number; $Q_n{}^t{}_{max}$ and $Q_n{}^t{}_{min}$ are the upper and lower outflow limits of the $n$th reservoir during the period $t$, respectively, in m$^3$/s. In this paper, the outflow constraint for each period is set according to the upper and lower outflow limits for each period and the reservoir operation rules in actual reservoir operation.

### 2.2.5. Output Constraint

The reservoir output is limited by the water level of reservoir, the maximum power of the hydraulic turbine, the maximum flow through turbine, and other engineering parameters. The output of the reservoir at any moment during the period $t$ should meet the following conditions:

$$N_n{}^t{}_{\min} \le N_n^t \le N_n{}^t{}_{\max} \tag{9}$$

where $t$ is the number of periods; $n$ is the reservoir number; $N_n^t$ is the average output of the $n$th reservoir during the period $t$ in kW; $N_n{}^t{}_{max}$ and $N_n{}^t{}_{min}$ are the upper and lower output limits of the $n$th reservoir during the period $t$, respectively, in kW.

### 3. Solution of the Multi-Objective Optimal Reservoir Operation Model

*3.1. Model Coding*

The optimal operation model of the cascade reservoirs takes a month as the regulation unit and a year as the operation unit. The model selects the periods from 31 October to 1 November of the following year, and the corresponding number of periods is $T = 12$. The decision variable selected for the model is the initial (final) water level $Z_i^t$ during the period $t$ of reservoir $i$ in cascade reservoirs, in which $i$ is the reservoir code and $t$ is the period code. The time dimension of the decision variable is $d_t = 12$, and the cascade reservoir group includes three reservoirs, namely Wanjiazhai, Sanmenxia, and Xiaolangdi. Therefore, the spatial dimension of the decision variable is $d_i = 3$, and the dimension of the decision variable matrix is $3 \times 12$.

In summary, the decision variable of the model is expressed as:

$$Z = \begin{bmatrix} Z_1^1, Z_1^2, \cdots, Z_1^t, \cdots, Z_1^T \\ Z_2^1, Z_2^2, \cdots, Z_2^t, \cdots, Z_2^T \\ Z_3^1, Z_3^2, \cdots, Z_3^t, \cdots, Z_3^T \end{bmatrix} \tag{10}$$

*3.2. Computation Steps*

The fast non-dominated genetic algorithm (NSGA-II) is a multi-objective optimization algorithm based on a genetic algorithm and combining non-dominated sorting and crowding distance sorting. A fast, non-dominated sorting genetic algorithm based on a successive approximation approach, namely SA-NSGA-II, which adds successive cycles, a variable search space of decision variables, and the selection of a Pareto optimal solution set relative to NSGA-II, is introduced in this study. The main parameters and process are described in detail below.

To set the number of cycles, the size of the maximum number of iterations, *generation*, of a single cycle in successive cycles is related to the total number of successive cycles, *K*, and the total number of iterations of cycles, max *run*. The number of iterations of a single cycle is variable with the following values:

$$generation^{(k)} = gs + \frac{k-1}{K-1}(gs - gf)$$
$$gf = 2\frac{\max run}{K} - gs \tag{11}$$
$$s.t.\, gf > gs$$

where $k$ is the number of cycles, $generation^{(k)}$ is the maximum number of iterations for the $k$th cycle, $gs$ is the maximum number of iterations for the initial cycle, and $gf$ is the maximum number of iterations for the last cycle. The maximum number of iterations of successive cycles increases in turn to continuously improve the optimizing ability of the current cycle. The specific values of the total number of iterations of cycles, max *run*, the total number of successive cycles, *K*, and the maximum number of iterations for the initial cycle, $gs$, depend on the complexity of the problem.

For the determination of the variable search space, the upper limit $X_{\max}$ and lower limit $X_{\min}$ of the search space of the decision variables change with the number of cycles, $k$, of successive cycles, i.e., the search scope decreases continuously at each stage:

$$k: \begin{array}{l} width^{(k)} = \frac{X_{\max} - X_{\min}}{ek} \\ X_{\max} = \left\{ x^t{}_{\max n} + width \right\} \\ X_{\min} = \left\{ x^t{}_{\min n} - width \right\} \\ s.t. \begin{cases} X_{\max} \leq X_{\max 1} \\ X_{\min} \geq X_{\min 1} \end{cases} \end{array} \tag{12}$$

where $k$ is the number of cycles, $n$ is the number of stages, $X_{max1}$ and $X_{min1}$ are the upper and lower limits of the search space of the decision variable at the beginning of the first cycle, respectively, and $e$ is the amplification coefficient. According to the size of the search space, the scaling controls the size of *width* to be adapted to the search scope. $X_{max}$ and $X_{min}$ are the upper and lower limits of the search space of the decision variables of the $k$th cycle, while $x^t_{maxn}$ and $x^t_{minn}$ are the upper and lower limits of the optimal solution set at the stage $n$ corresponding to the Pareto optimal solution set obtained in the $(k-1)$th cycle.

The population size has a large impact on the optimization results. If it is too small, the global search ability will be poor and finding the optimal solution may be difficult. If it is too large, processing will take longer and have a lower efficiency. The specific value can be determined based on the size of the search space. In order to ensure the search efficiency with certain precision, the number of populations can be adjusted by the following equation:

$$k : popsize = \max pop - \frac{(k-1)(\max pop - \min pop)}{\max run} \tag{13}$$

where $k$ is the number of cycles, *popsize* is the population size of the $k$th cycle, max *pop* is the population size with the maximum number of populations in the first cycle, and min *pop* is the population size with the minimum number of populations in the last cycle.

To select the Pareto optimal solution set, the total number of successive cycles is $K$. In every cycle, the same Pareto optimal solution set as the number of populations is generated, and the Pareto optimal solution sets of all single cycles are mixed by non-dominated and crowding distance sorting, among which the non-dominated solution set is selected as the final Pareto optimal solution set of the algorithm.

In addition, the setting of parameters such as the cross distribution and variance distribution index in the algorithm is no different from the NSGA-II algorithm [49]. The overall program block diagram of SA-NSGA-II is shown in Figure 1.

To determine the decision variables of the model, the monthly average inflows of the Wanjiazhai reservoir from 31 October to 1 November of the following year in typical years (high flow/sediment year, median water/sediment year, and low flow/sediment year) are selected as input parameters, respectively. The model is solved, with specific steps as follows.

Step 1: The model parameters are initialized, and the population size $popsize = N$ the total number of cycles $K$, and the number of iterations of single cycle gen are set. The inflow of upstream reservoir, operation cycle, initial water level for regulation, interval inflow, and downstream suitable ecological flow are input. The constraints are set and the decision variables according to Equation (7) with its upper and lower limits $Z_{min}$ and $Z_{max}$ of the search space are generated.

Step 2: Based on the upper and lower limits $Z_{min}$ and $Z_{min}$ of the current search space, the decision variables for each reservoir during each period are randomly generated, which constitute the initial population.

Step 3: The decision variables for each reservoir during each period of the current population must comply with the constraints. Accordingly the decision variables that violate the constraints must be rectified, and the target values for each individual in the population, i.e., $M_1$ (average output during the period of operation) and $M_2$ (average change in downstream ecological flow during the period of operation), are calculated.

Step 4: Simulated binary crossover and polynomial variation on the current parent $P$ are performed to generate the offspring $Q$. Step 3 is performed on the offspring $Q$.

Step 5: The parent $P$ and the offspring $Q$ are mixed to produce a mixed population $R = P \cup Q$, and the mixed population R is sorted non-dominantly to generate each non-dominant layer $(F_1, F_2, \cdots, F_L)$.

Step 6: The crowding distance of each non-dominated layer is sorted, and individuals to be incorporated into the next generation of parent population according to the sorting result are selected until the number of populations reaches $N$ and the next generation of parent population is generated.

Step 7: If the current iteration termination condition is met, continue with Step 8; if not, return to Step 4.

Step 8: The current population $P$ is saved and the decision scheme is set to $K = 5$. If the total cycle termination condition is satisfied, the Pareto optimal solution set of decision solution set $S$ is output as the set of cascade reservoir operation solution; if not, generate a new search space and return to Step 2.

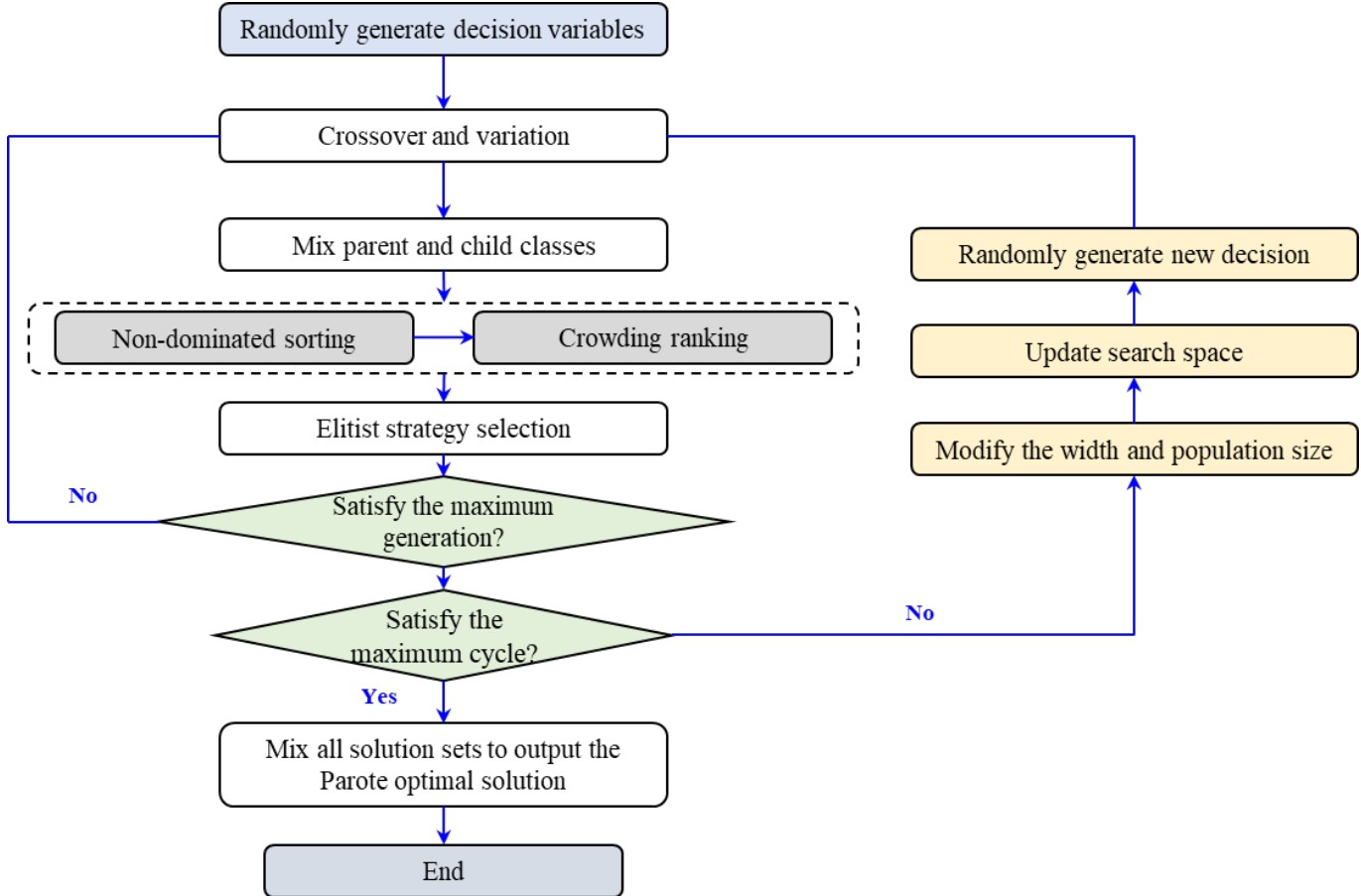

**Figure 1.** Overall program block diagram of SA-NSGA-II.

## 4. Analysis of the Multi-Objective Operation Results

The optimal operation values of the cascade reservoir group in the middle reaches of the Yellow River in three kinds of typical year (high flow/sediment year, median water/sediment year, and low flow/sediment year) were selected. (The data selection time range is from 30 April 2006 to 31 December 2022). Among them, November 2008 to October 2009 was selected for the high flow/sediment year, November 2010 to November 2011 was selected for the median water/sediment year, and November 2016 to October 2017 was selected for the low flow/sediment year. The relevant data of the Wanjiazhai, Sanmenxia, and Xiaolangdi reservoirs were obtained from the hydrological yearbook of the Yellow River basin, as shown in Table 1, in which the suitable ecological flow of the downstream river during each period (month) of each reservoir was calculated by the monthly frequency computation method.

SA-NSGA-II was used to optimize the actual cascade reservoir model for solution. The specific parameters of the algorithm were set as follows: the total number of cycles $K = 5$, the number of populations $\max pop = 250$ and $\min pop = 200$, the number of iterations of a single cycle $gs = 200$, the cross distribution index of the algorithm is 20, and the variance distribution index is 20. According to the solution process, the Pareto optimal solution sets of the two-objective optimal operation model of the cascade reservoir group in the

middle Yellow River for each typical year are computed by inputting the inflow data of the corresponding years.

**Table 1.** The mean monthly discharge, upper limit of water level, and lower limit of water level in the Wanjiazhai, Sanmenxia, and Xiaolangdi reservoirs.

| Typical Year | Years with High Flow/Sediment (November 2008–October 2009) | | | Years with Median Flow/Sediment (November 2010–October 2011) | | | Years with Low Flow/Sediment (November 2016–October 2017) | | |
|---|---|---|---|---|---|---|---|---|---|
| Mean Monthly Discharge (m³/s) | Wanjiazhai | Sanmenxia | Xiaolangdi | Wanjiazhai | Sanmenxia | Xiaolangdi | Wanjiazhai | Sanmenxia | Xiaolangdi |
| 11 | 515 | 734 | 758 | 451 | 558 | 504 | 289 | 458 | 442 |
| 12 | 249 | 313 | 336 | 360 | 546 | 444 | 392 | 528 | 494 |
| 1 | 290 | 326 | 310 | 376 | 429 | 322 | 362 | 453 | 419 |
| 2 | 403 | 591 | 658 | 440 | 506 | 482 | 445 | 445 | 444 |
| 3 | 954 | 916 | 984 | 768 | 866 | 874 | 546 | 614 | 539 |
| 4 | 990 | 918 | 980 | 700 | 680 | 644 | 301 | 579 | 529 |
| 5 | 305 | 443 | 508 | 347 | 489 | 440 | 192 | 298 | 283 |
| 6 | 350 | 494 | 663 | 392 | 504 | 468 | 217 | 402 | 391 |
| 7 | 336 | 383 | 391 | 406 | 599 | 670 | 321 | 464 | 542 |
| 8 | 504 | 640 | 625 | 447 | 744 | 668 | 354 | 637 | 625 |
| 9 | 1063 | 1484 | 1577 | 915 | 2460 | 2486 | 716 | 1020 | 1079 |
| 10 | 521 | 712 | 632 | 487 | 961 | 935 | 517 | 1228 | 1087 |
| | Wanjiazhai | | | Sanmenxia | | | Xiaolangdi | | |
| Maximum Water Level (m) [a] | 980.01 | | | 319.42 | | | 273.5 | | |
| Minimum Water Level (m) | 921.38 | | | 283.46 | | | 205.01 | | |

Note: [a] Water level data selected between 30 April 2006 and 31 December 2022.

### 4.1. High Flow/Sediment Year

The number of Pareto optimal solution sets obtained by solving the two-objective optimal operation model for the cascade reservoir group in the typical high flow/sediment year was 138, i.e., 138 optimal operation schemes for the cascade reservoir group, as shown in Table 2. The maximum average output of the target value of the operation scheme set A was $3.2013 \times 10^5$ kW, and the minimum average output was $2.8759 \times 10^5$ kW. The maximum average change in ecological flow was 0.1389, and the minimum was 0.0745.

The Pareto frontier distribution of reservoir operation schemes in the target space is shown in Figure 2. There is a negative correlation between the two objectives of each operation scheme in the target space. Each frontier point is relatively evenly distributed.

**Table 2.** Operation scheme set in the typical year of high flow/sediment.

| Operation Scheme Set A | Average Output ($10^5$ kW) | Average Change in Ecological Flow |
|---|---|---|
| A1 | 2.8759 | 0.0745 |
| A2 | 2.8886 | 0.0752 |
| . . . | . . . | . . . |
| A138 | 3.2013 | 0.1389 |

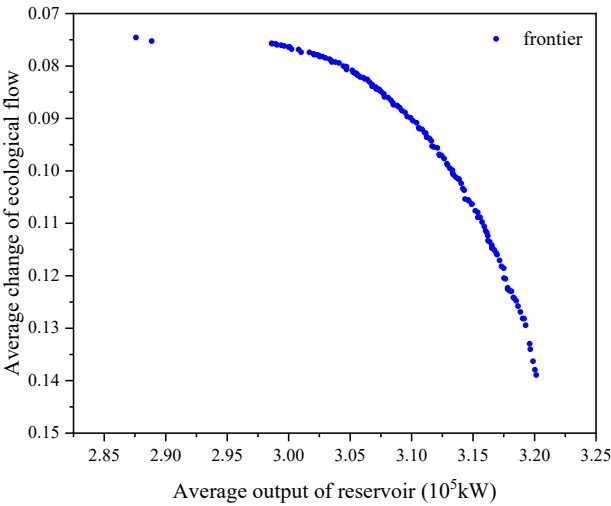

**Figure 2.** Pareto frontier of the operation scheme for a high flow/sediment year.

*4.2. Median Water/Sediment Year*

The number of Pareto optimal solution sets obtained by solving the two-objective optimal operation model for the cascade reservoir group in the typical year of median water/sediment is 111, i.e., 111 operation schemes for the cascade reservoir group.

As shown in Table 3, the maximum average output of the target value of the operation scheme set B is $2.8324 \times 10^5$ kW, and the minimum average output is $2.6493 \times 10^5$ kW. The maximum average change in ecological flow is 0.2627, and the minimum is 0.1908. The Pareto frontier distribution of the reservoir operation scheme in the target space is shown in Figure 3. The Pareto frontier changes are comparable with the previous typical year. The change in incoming water and sediment from the upstream results in changes in hydraulic and constraining relationships among reservoirs, affecting the number and distribution of solution sets.

**Table 3.** Operation scheme set in the typical year of median water/sediment.

| Operation Scheme Set B | Average Output ($10^5$ kW) | Average Change in Ecological Flow |
|:---:|:---:|:---:|
| B1 | 2.6493 | 0.1908 |
| B2 | 2.6496 | 0.1910 |
| . . . | . . . | . . . |
| B111 | 2.8210 | 0.2502 |

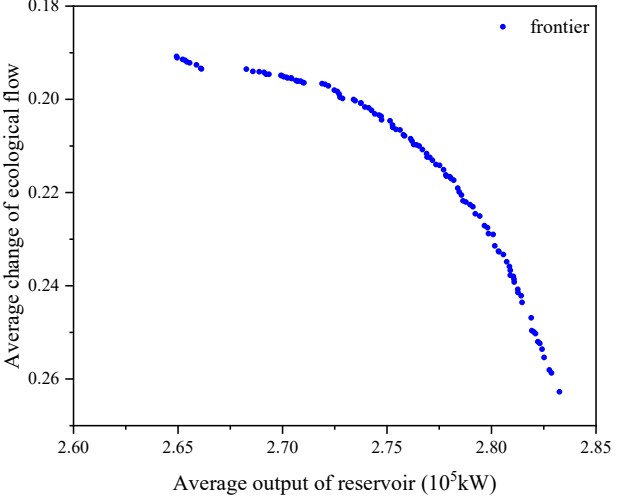

**Figure 3.** Pareto frontier of the operation scheme for a median water/sediment year.

*4.3. Low Flow/Sediment Year*

The number of Pareto optimal solution sets obtained by solving the two-objective optimal operation model for the cascade reservoir group in a typical low flow/sediment year is 91, i.e., 91 operation schemes for the cascade reservoir group.

Table 4 shows that the maximum average output of the target value of the operation scheme set C is $2.5702 \times 10^5$ kW, and the minimum average output is $2.4513 \times 10^5$ kW. The maximum average change in the ecological flow is 0.2892, and the minimum is 0.2130.

**Table 4.** Operation scheme set in the typical year of low flow/sediment.

| Operation Scheme Set C | Average Output ($10^5$ kW) | Average Change in Ecological Flow |
|:---:|:---:|:---:|
| C1 | 2.5540 | 0.2691 |
| C2 | 2.5552 | 0.2705 |
| . . . | . . . | . . . |
| C91 | 2.5518 | 0.2653 |

The Pareto front distribution of reservoir operation schemes in the target space is shown in Figure 4. Based on Figures 2–4, we conclude that the upper and lower limits of the range of target value for the average output of the operation schemes are decreasing in parallel with the reducing incoming water and sediment, and the average change in ecological flow is also affected.

From the above results, it can be shown that the average output and ecological flow changes in reservoirs in years with a high flow/sediment, median flow/sediment, and low flow/sediment generally exhibit a competitive relationship. When reservoirs pursue power generation benefits, they will sacrifice some ecological benefits. Moreover, this competitive relationship is closely related to the water volume, such as significant differences in overall power generation efficiency in years with a high flow/sediment, median flow/sediment, and low flow/sediment, ranging from 2.8759 to $3.2014 \times 10^5$ kW, 2.6493 to $2.8267 \times 10^5$ kW, and 2.5540 to $2.5518 \times 10^5$ kW, respectively. The ecological benefits have the same pattern, with the average change range of the three typical annual ecological flows being 0.0746–0.1389, 0.1908–0.2587, and 0.2691–0.2653, respectively. The overall ecological benefits are still closely related to annual runoff, and there is more water available for regulation in a high flow/sediment year, so the degree of regulation is also the highest, followed by a median flow/sediment year, and the worst in low flow/sediment years. Overall, annual runoff is the most important factor affecting power generation and ecological benefits. The more water there is, the greater the degree of regulation and the greater the benefits generated. However, due to the competitive relationship between power generation efficiency and ecological benefits, there is a trend of eliminating each other. Therefore, how to scientifically select the most suitable scheduling plan from the optimal solution set is worth in-depth research.

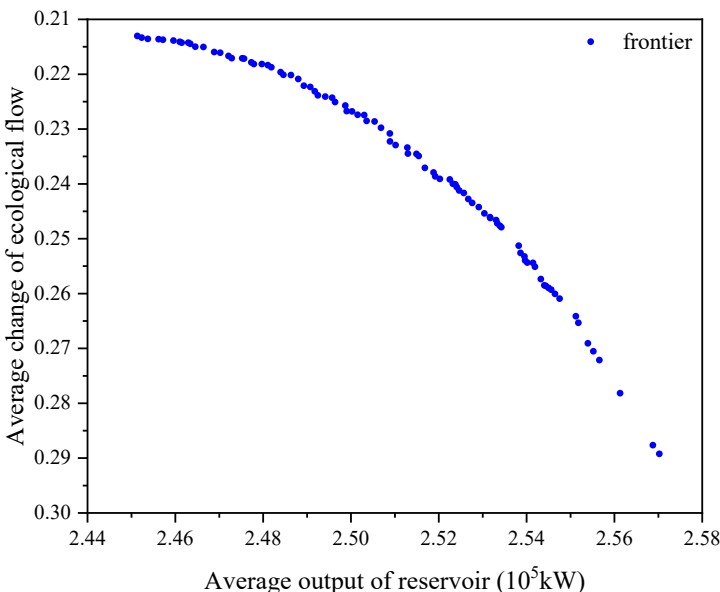

**Figure 4.** Pareto frontier of the operation scheme for a low flow/sediment year.

## 5. Comprehensive Evaluation of the Multidimensional Coordination Operation Scheme of Cascade Reservoirs

The cascade reservoir operation scheme for the Wanjiazhai. Sanmenxia, and Xiaolangdi reservoirs was obtained based on the multi-objective algorithm. Each scheme in the set was a non-dominated solution of the operation model, and the optimal solution meets the multi-objective operation model. In order to determine the operation scheme with the greatest comprehensive benefits in each typical year, the multi-objective reservoir operation scheme needed to be evaluated and optimized. This paper constructs an evaluation index system for reservoir operation schemes, applies a combined approach of objective and subjective evaluation, and introduces the gray target and cumulative prospect theories. Moreover, this paper quantifies the established scheme evaluation index system uniformly and optimizes the reservoir operation scheme by the fuzzy evaluation method to select the recommended scheme for each typical year, which provides a new scientific formulation of reservoir operation schemes in the middle reaches of the Yellow River.

### 5.1. Evaluation Index System

To further distinguish the advantages and disadvantages of the comprehensive benefits of each optimal reservoir operation scheme set, an evaluation index system is constructed according to the target space of each optimal operation scheme. This system mainly includes three perspectives: power generation, ecological, and flood control indices. Specifically, in the optimal operation scheme of the cascade reservoir group, the first-level index consists of three second-level indices. The second-level indices $X_{k1}$, $X_{k2}$, and $X_{k3}$ correspond to the average output of the reservoir, the average change in downstream ecological flow of the reservoir, and the maximum peak clipping rate of the reservoir, respectively, as shown in Figure 5.

For the optimal operation scheme for single reservoirs, there is only one second-level index under the first-level one, namely the optimal operation scheme target of the single reservoir.

(1) The power generation index $X_1$ is composed of the average output index, comprising the second-level index of each reservoir during the corresponding operation period. When the average output is higher, the benefits of the reservoir are greater. The average output index is defined as an income-oriented index.

(2) The ecological index $X_2$ is composed of the average change index, comprising the second-level index of downstream ecological flow during the corresponding operation period of each reservoir. When the average change in ecological flow is greater, the

ecological benefits of the reservoir are reduced. The ecological flow change degree index is defined as a cost-oriented index.

(3) The flood control index $X_3$ is comprised of the maximum peak clipping rate index, encompassing the second-level index of each reservoir during the corresponding operation period. When the maximum peak clipping rate is larger, the benefit of flood control of the corresponding reservoir is greater. The maximum peak clipping rate index is defined as a revenue-oriented index. For cascade reservoirs, the maximum peak clipping rate can be calculated according to the inflow and outflow of each reservoir during the operation period.

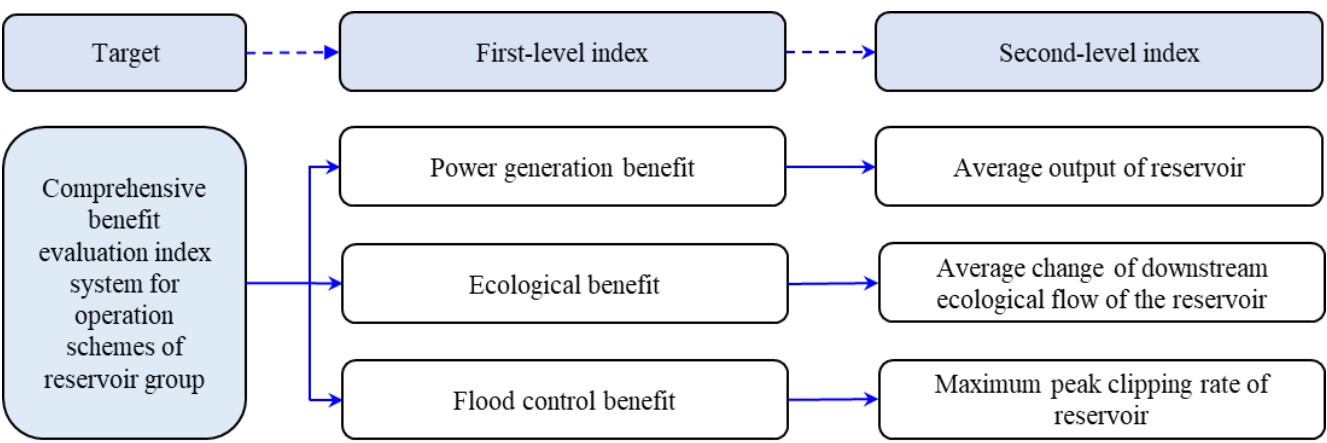

**Figure 5.** Comprehensive benefit evaluation index system of operation schemes of the reservoir group.

*5.2. Evaluation Index Method*

For optimal reservoir operation, various factors need to be considered. There are uncertain relations among these various factors. The gray target theory is a gray correlation analysis theory proposed by Deng Julong [50] that utilizes data indicators that best reflect the advantages and disadvantages of the scheme to form corresponding data patterns.

The cumulative prospect theory is an improvement on prospect theory [51]. The cumulative prospect theory achieves comprehensive analysis of multiple results, allowing different weight functions for gains and losses. The size of a prospect value is determined by the value function and decision weight together. Different value and weight functions are selected by setting a reference point according to the gain or loss determined by the data indicators. In this evaluation system, the value function represents the quantification of the evaluation index system for risk aversion or pursuit states in the face of gain or loss. The weight function is used to balance the structural proportion of each index in the evaluation system and seeks to maximize the prospect value.

*5.3. Evaluation Index System Data*

Based on the optimal operation scheme set of the Wanjiazhai, Sanmenxia, and Xiaolangdi cascade reservoir group, the power generation, ecological, and flood control indices of each reservoir are calculated. The initial values of the second-level index for each typical year's cascade reservoir operation scheme are shown in Tables 5–7.

**Table 5.** Index data of the cascade reservoir operation scheme in years with a high flow/sediment.

| Operation Scheme Set A | Power Generation Index $X_1$ | | | Ecological Index $X_2$ | | | Flood Control Index $X_3$ | | |
|---|---|---|---|---|---|---|---|---|---|
| | $X_{11}$ | $X_{12}$ | $X_{13}$ | $X_{21}$ | $X_{22}$ | $X_{23}$ | $X_{31}$ | $X_{32}$ | $X_{33}$ |
| A1 | 3.3946 | 0.0929 | 0.1431 | 1.9825 | 0.0913 | 0.2343 | 3.2503 | 0.0395 | 0.4922 |
| A2 | 3.3936 | 0.0945 | 0.1442 | 1.9988 | 0.0907 | 0.2356 | 3.2732 | 0.0405 | 0.4907 |
| ... | ... | ... | ... | ... | ... | ... | ... | ... | ... |
| A138 | 3.4746 | 0.0977 | 0.1008 | 2.2681 | 0.0970 | 0.2506 | 3.8612 | 0.2220 | 0.9874 |

**Table 6.** Index data of the cascade reservoir operation scheme in years with a median flow/sediment.

| Operation Scheme Set B | Power Generation Index $X_1$ | | | Ecological Index $X_2$ | | | Flood Control Index $X_3$ | | |
|---|---|---|---|---|---|---|---|---|---|
| | $X_{11}$ | $X_{12}$ | $X_{13}$ | $X_{21}$ | $X_{22}$ | $X_{23}$ | $X_{31}$ | $X_{32}$ | $X_{33}$ |
| B1 | 2.9483 | 0.0941 | 0.1380 | 1.8842 | 0.4029 | 0.2913 | 3.1153 | 0.0753 | 0.8529 |
| B2 | 2.9473 | 0.0939 | 0.1376 | 1.8843 | 0.4028 | 0.2856 | 3.1171 | 0.0764 | 0.8583 |
| ... | ... | ... | ... | ... | ... | ... | ... | ... | ... |
| B111 | 3.0063 | 0.1105 | 0.1055 | 2.0555 | 0.3961 | 0.3706 | 3.4241 | 0.2694 | 0.8938 |

**Table 7.** Index data of the cascade reservoir operation scheme in years with a low flow/sediment.

| Operation Scheme Set C | Power Generation Index $X_1$ | | | Ecological Index $X_2$ | | | Flood Control Index $X_3$ | | |
|---|---|---|---|---|---|---|---|---|---|
| | $X_{11}$ | $X_{12}$ | $X_{13}$ | $X_{21}$ | $X_{22}$ | $X_{23}$ | $X_{31}$ | $X_{32}$ | $X_{33}$ |
| C1 | 2.2495 | 0.1630 | 0.1615 | 1.9110 | 0.4174 | 0.3133 | 3.5013 | 0.2266 | 0.9721 |
| C2 | 2.2503 | 0.1632 | 0.1608 | 1.9130 | 0.4187 | 0.3159 | 3.5023 | 0.2295 | 0.9607 |
| ... | ... | ... | ... | ... | ... | ... | ... | ... | ... |
| C91 | 2.2435 | 0.1516 | 0.1801 | 1.9220 | 0.4202 | 0.2709 | 3.4897 | 0.2240 | 0.9872 |

*5.4. Optimization and Analysis of Reservoir Operation Scheme*

Taking the optimal selection of the cascade reservoir operation scheme for the Wanji-azhai, Sanmenxia, and Xiaolangdi reservoirs in typical years with a high flow/sediment as an example, the average output of the three reservoirs in the comprehensive benefit evaluation index system under the power generation index are benefit-oriented indexes. The decision matrix of the power generation index is standardized to determine the positive (negative) bullseye of each second-level index and obtain the positive (negative) bullseye coefficient matrix from the power generation index, as shown in Table 8.

Similarly, the positive (negative) bullseye coefficient matrix of the ecological and flood control index can be calculated in sequence. Next, the comprehensive prospect values of each first-level index are calculated according to the cumulative prospect theory. Firstly, the index weights of the second-level index are set. As the main controlling reservoir in the middle reaches of the Yellow River, the Xiaolangdi reservoir has a corresponding index weight of 0.4. The Wanjiazhai and Sanmenxia reservoirs, as supplementary reservoirs in the middle reaches of the Yellow River, have corresponding index weights of 0.3. Secondly, the comprehensive prospect values of the power generation, ecological, and flood control indices are calculated, as shown in Table 9.

According to Table 9, based on the comprehensive prospect values of each first-level index, the fuzzy evaluation method is used to evaluate the comprehensive benefit value of the reservoir operation scheme set. The fuzzy evaluation matrix of each first-level index was calculated and established, and the membership degree of sediment regulation potential for each water conservancy hub under five evaluation levels was calculated, as shown in

Table 10. Among them, Scheme A138 has a greater membership degree than other schemes at Grade 1 (very high), 2 (high), and 3 (general), and lower than them at Grade 4 (low) and 5 (very low), indicating that the comprehensive benefit of this scheme is better than that of other schemes. The comprehensive benefit scores and rankings of all operation schemes are shown in Table 11. Among them, Scheme A138 has the highest score (73.2936) and ranks first, while Scheme A43 has the lowest score (68.6195) and ranks 138th. The optimal operation scheme for the cascade reservoirs in each typical year is shown in Table 12, and the corresponding actual operation data are shown in Table 13.

**Table 8.** Positive and negative bullseye coefficients of the power generation index.

| Scheme Set A No. | Positive Bullseye Coefficient | | | Negative Bullseye Coefficient | | |
|---|---|---|---|---|---|---|
| | $X_{11}$ | $X_{12}$ | $X_{13}$ | $X_{11}$ | $X_{12}$ | $X_{13}$ |
| Scheme A1 | 0.3362 | 0.3333 | 0.3333 | 0.9752 | 1 | 1 |
| Scheme A2 | 0.3333 | 0.3464 | 0.3419 | 1 | 0.8980 | 0.9301 |
| ... | ... | ... | ... | ... | ... | ... |
| Scheme A138 | 0.9723 | 0.9893 | 1 | 0.3365 | 0.3345 | 0.3333 |

**Table 9.** Comprehensive prospect values of first-level indexes.

| Scheme Set A No. | $X_1$ | $X_2$ | $X_3$ |
|---|---|---|---|
| Scheme A1 | −1.6355 | 0.0664 | −0.7731 |
| Scheme A2 | −1.5573 | −0.1248 | −0.6834 |
| ... | ... | ... | ... |
| Scheme A138 | 0.6565 | −1.5664 | −0.1283 |

**Table 10.** Membership degree under each evaluation level.

| Scheme Set A No. | Very High | High | General | Low | Very Low |
|---|---|---|---|---|---|
| Scheme A1 | 0.1016 | 0.1482 | 0.2001 | 0.2520 | 0.2979 |
| Scheme A2 | 0.0859 | 0.1350 | 0.1953 | 0.2608 | 0.3229 |
| ... | ... | ... | ... | ... | ... |
| Scheme A138 | 0.1553 | 0.1876 | 0.2112 | 0.2230 | 0.2229 |

**Table 11.** Comprehensive benefit score of the scheme.

| Scheme Set A No. | Score | Ranking |
|---|---|---|
| Scheme A1 | 70.0368 | 45 |
| Scheme A2 | 69.0019 | 94 |
| ... | ... | ... |
| Scheme A138 | 73.2936 | 1 |

**Table 12.** Optimal operation scheme of cascade reservoirs of the Wanjiazhai, Sanmenxia, and Xiaolangdi reservoirs.

| Typical Year | Scheme No. | Average Output /$10^5$ kW | Average Change in Ecological Flow | Score |
|---|---|---|---|---|
| Year with high flow/sediment | Scheme A138 | 3.2013 | 0.1389 | 73.29 |
| Year with median flow/sediment | Scheme B66 | 2.8324 | 0.2627 | 73.17 |
| Year with low flow/sediment | Scheme C33 | 2.4513 | 0.2130 | 72.51 |

**Table 13.** Actual operation data of cascade reservoirs of the Wanjiazhai, Sanmenxia, and Xiaolangdi reservoirs.

| Typical Year | Average Output/$10^5$ kW | Average Change in Ecological Flow |
|---|---|---|
| Years with high flow/sediment | 2.8245 | 0.1575 |
| Years with median flow/sediment | 2.5656 | 0.3703 |
| Years with low flow/sediment | 2.1067 | 0.3803 |

Table 12 shows the operation scheme set of the cascade reservoirs, which is optimized based on the gray target-cumulative prospect theory and the fuzzy evaluation method. Among them, the optimal operation schemes corresponding to the typical years with a high and median flow/sediment feature the maximum average output in the scheme set but the most average change in ecological flow. Since the evaluation method is oriented by loss aversion, Schemes A138 and B66 have the highest returns in terms of the power generation index and comparatively low losses in the ecological index during large inflows of water and sediment. The optimal scheme corresponding to a typical year with a low flow/sediment is characterized by a minimal average change in ecological flow. In the case of a small inflow of water and sediment, the optimal scheme C33 is given priority to ensure ecological benefits while taking power generation benefits into account.

Table 13 shows a further comparison based on actual operation data for cascade reservoirs for each typical year, in which Scheme A138 intersects the actual operation data, increasing the average output by 13.34% and reducing the average change in ecological flow by 11.81%. Scheme B66 intersects with the actual operation data, increasing the average output by 10.40% and reducing the average change in ecological flow by 29.05%. Scheme C33 intersects with the actual operation, increasing the average output by 16.36% and reducing the average change degree in ecological flow by 43.99%. Since the multi-objective joint optimal operation of the Wanjiazhai, Sanmenxia, and Xiaolangdi reservoirs utilizes the unified management of the operation data of each reservoir, such as flow and water level operation, and the optimization algorithm has continuously optimized each operation target, it allows the reasonable allocation and regulation of water resources in the middle reaches of the Yellow River. Therefore, compared with the actual data of the reservoirs in the middle reaches based on their respective operation rules, the optimized operation plans have great advantages in terms of the comprehensive benefits, such as average output and average change in ecological flow.

The dynamic water levels of each reservoir in each optimal operation scheme during the operation period are shown in Figure 6.

The proposed evaluation decision-making method based on gray target theory and cumulative prospect theory-fuzzy evaluation method, which introduces gray target theory and cumulative prospect theory, uniformly quantifies the established scheme evaluation index system, and can effectively deal with uncertain issues such as cognitive limitations and bounded rationality of decision-makers. The fuzzy evaluation method is used to optimize the reservoir operation plan, reducing the subjectivity of the decision-making process as much as possible. This method can comprehensively consider the impact of different weighting methods and decision-maker bounded rationality on the evaluation results, and the recommended scheme can better reflect the multi-objective scheduling purpose, resulting in more reasonable results.

In addition to flood control and sedimentation reduction during the flood season, the reservoirs in the Yellow River basin, along with other reservoirs in the upper reaches of the Yellow River, also undertake various scheduling tasks such as ecology, water supply, and irrigation. Selecting a relatively simple objective function for comprehensive power generation benefits and ecological benefits can be applicable to the operation of cascade reservoirs during non-flood seasons; however, for the Yellow River, which is severely short of water resources, with severe soil erosion, and frequent downstream water disasters, a

multi-reservoir joint scheduling approach should be adopted to establish a multi-objective scheduling model for reservoir groups that comprehensively considers flood control, power generation, sediment reduction, ecology, and water supply irrigation, and explore its efficient solution methods. Relevant research work will be carried out in the future.

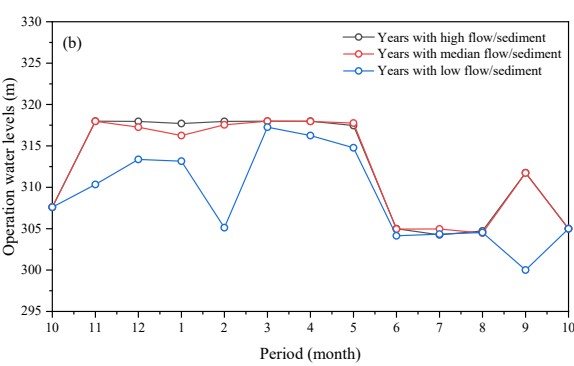

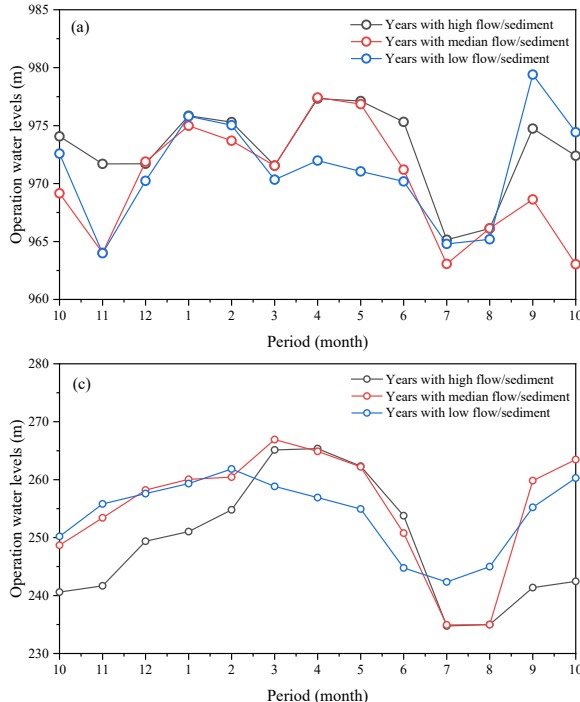

**Figure 6.** Water level process of the optimal operation schemes. (**a**) Wanjiazhai Reservoir, (**b**) Sanmenxia Reservoir, (**c**) Xiaolangdi Reservoir.

## 6. Conclusions

(1) SA-NSGA-II, a two-objective optimal operation model of the Wanjiazhai, Sanmenxia, and Xiaolangdi cascade reservoir group in the middle reaches of the Yellow River, is solved for three typical years, with multiple Pareto optimal operation sets and formulating three reservoir operation scheme sets, including 138, 111, and 91 operation schemes, respectively. The operation scheme set fully utilizes the water resources of the middle reaches of the Yellow River by optimizing the ecological benefits of the downstream river channel, while ensuring the output of the reservoir.

(2) This paper illustrates the construction of a comprehensive benefit evaluation index system for reservoir operation schemes, based on the three first-level indices of power generation, ecology, and flood control. This is achieved by integrating each reservoir's data and combining the gray target theory and cumulative prospect theories to obtain recommended schemes. The selected schemes are similar to A138, B66, and C33 in high flow/sediment, median flow/sediment, and low flow/sediment years, respectively.

(3) Through evaluation and optimization, six optimal schemes were obtained for each typical year. Compared with actual operation data, Scheme A138 increases the average output by 13.34% compared to actual operation, and decreases the average change in ecological flow by 11.81%; Scheme B66 intersects with actual scheduling, increasing the average output by 10.40% and reducing the average change in ecological flow by 29.05%; Scheme C33 intersects with actual scheduling, increasing the average output by 16.36% and reducing the average change in ecological flow by 43.99%, which can simultaneously balance power generation and ecological benefits. This provides a new decision-making approach and technical support for the optimal operation of reservoirs in the middle reaches of the Yellow River.

**Author Contributions:** Conceptualization, X.L.; methodology, Q.W. and D.L.; software, J.L.; validation, Q.W. and Y.W.; formal analysis, Q.W., X.L. and Y.W.; investigation, J.L. and D.L.; resources, X.L. and Y.W.; data curation, H.Z. and J.L.; writing—original draft preparation, X.L.; writing—review and editing, Q.W., J.L. and D.L.; visualization, H.Z. and D.L.; supervision, Q.W.; project administration, Y.W.; funding acquisition, X.L., Q.W. and Y.W. All authors have read and agreed to the published version of the manuscript.

**Funding:** This research was supported by the National Natural Science Foundation of China (Grant No. U2243236, U2243215, 52309091, 51879115), the Science and Technology Program of Henan Province (Grant No. 222300420235), Major Science and Technology Project of the Ministry of Water Resources (SKS-2022088, SKR-2022021), and the basic research Fund Project of Yellow River Water Institute of Hydraulic Research (Grant No. HKY-JBYW-2022-12). The authors are grateful to the editors and anonymous reviewers for their insightful comments and suggestions.

**Data Availability Statement:** The original data presented in the current study are available by special request of the corresponding author.

**Acknowledgments:** The authors express sincere gratitude to Li Like and Yang Fei for their comments and suggestions on this article, as well as to the reviewers and editors.

**Conflicts of Interest:** The authors declare no conflict of interest.

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
