# Peer review of "Multi-Dimensional Collaborative Operation Model and Evaluation of Cascade Reservoirs in the Middle Reaches of the Yellow River"

_water, doi:10.3390/w15193523_

Round 1
Reviewer 1 Report
The paper mainly focuses on the establishment of a two-objective optimal operation model for the reservoir group in the middle reaches of the Yellow River, which aims to maximize the average output of the cascade reservoir group and minimize the average change of ecological flow during the operation period under efficient sediment transport conditions, with the coordination degree of water and sediment as the constraints of reservoir discharge flows.
The paper also constructs an evaluation index system for reservoir operation schemes, applies a combined approach of objective and subjective evaluations, and introduces the gray target and cumulative prospect theories. By uniformly quantifying the established scheme evaluation index system, screening the reservoir operation schemes with the fuzzy evaluation method, and selecting the recommended scheme for each typical year, the paper provides a new scientific formulation of the operation schemes of reservoirs in the middle reaches of the Yellow River. The selected schemes are compared with actual data, demonstrating the effectiveness of joint reservoir operation and for multidimensional benefits in terms of power generation, ecology, and flood control.
The following questions need to be responded:
1.In the last paragraph of the introduction, it is mentioned that "In-depth research on the multi-objective optimal operation of water, sediment, electricity, and ecology of cascade reservoirs under variations in water and sediment is necessary." However, the research just mentioned what about ecology and electricity in its conclusion, which lacks enough theory support. Moreover, more recent work related to reservoir should be summarized: An integrated model with stable numerical methods for fractured underground gas storage. Journal of Cleaner Production, 2023, 393, 136268.
2.the transparency of data is incompetent, the paper just mentioned all the data from hydrological yearbook, but it didn’t list some real data which makes all the conclusion lack reliability, besides, the illustration of models is too simple, which make the optimization operate method difficult to repeat.
3.In section 5.4, it analyzed computation results "Scheme A138 intersects the actual operation data, increasing the average output by 13.34% and reducing the average change in ecological flow by 11.81%. Scheme B66 intersects with the actual operation data, increasing the average output by 10.40% and reducing the average change in ecological flow by 29.05%. Scheme C33 intersects with the actual operation, increasing the average output by 16.36% and reducing the average change degree in ecological flow by 43.99%". How to evaluate the reduction of average change in ecological flow and the increase of average output? Which scheme is the optimal scheme for this multi-objective optimal problem?
4.In the first paragraph of the introduction, it mentioned "reservoir construction can disrupt the natural equilibrium of the natural river flow structure and riverbed morphology". However, readers cannot find relationship between natural disruption and ecologic flow.
5.In figure 2. there are two points at the left of the picture. How to deal with these two points? Besides, figure 3. & figure 4. also exist some points away from point cluster. How to explain this phenomenon?
Author Response
- In the last paragraph of the introduction, it is mentioned that "In-depth research on the multi-objective optimal operation of water, sediment, electricity, and ecology of cascade reservoirs under variations in water and sediment is necessary." However, the research just mentioned what about ecology and electricity in its conclusion, which lacks enough theory support. Moreover, more recent work related to reservoir should be summarized: An integrated model with stable numerical methods for fractured underground gas storage. Journal of Cleaner Production, 2023, 393, 136268..
Authors’ reply: This article focuses on the collaborative scheduling between power generation and ecological benefits in the joint scheduling process of Wanjiazhai, Sanmenxia, and Xiaolangdi reservoirs in the Yellow River Basin. As stated by the reviewer, the regulation of sediment is not involved, so this sentence is unreasonable and has been revised. Secondly, this article also supplements the latest research on reservoirs, with specific modifications shown in lines 123-125.
“With the interdisciplinary research and application of different disciplines, the research results of underground gas storage can also provide a new perspective for reservoir scheduling[48].”
“[48]Xue, W., Wang, Y., Chen, Z., & Liu, H. An integrated model with stable numerical methods for fractured underground gas storage. Journal of Cleaner Production, 2023, 393, 136268.”
- the transparency of data is incompetent, the paper just mentioned all the data from hydrological yearbook, but it didn’t list some real data which makes all the conclusion lack reliability, besides, the illustration of models is too simple, which make the optimization operate method difficult to repeat.
Authors’ reply: The author supplemented the input conditions of the model, including the flow process of the selected typical year and the upper and lower boundary conditions of the water level, as shown in Table 1. For the construction of the model, this article provides a detailed introduction to the objective function, constraint conditions, and calculation steps of the model. However, due to the lack of relevant data on flow and water level, the operability of the article is not strong, and the author has supplemented it. Secondly, due to the relatively mature algorithm of multi-objective scheduling, this article uses a citation method for explanation, without a detailed introduction to the algorithm. Readers can refer to the cited literature, The specific supplementary boundary conditions are as follows:
“Table 1. The mean monthly discharge, upper limit of water level, and lower limit of water level in Wanjiazhai, Sanmenxia, and Xiaolangdi reservoirs
|
Typical year |
Years with high flow/sediment (2008.11-2009.10) |
Years with median flow/sediment (2010.11-2011.10) |
Years with low flow/sediment (2016.11-2017.10) |
||||||
|
mean monthly discharge(m3/s) |
Wanjiazhai |
Sanmenxia |
Xiaolangdi |
Wanjiazhai |
Sanmenxia |
Xiaolangdi |
Wanjiazhai |
Sanmenxia |
Xiaolangdi |
|
11 |
515 |
734 |
758 |
451 |
558 |
504 |
289 |
458 |
442 |
|
12 |
249 |
313 |
336 |
360 |
546 |
444 |
392 |
528 |
494 |
|
1 |
290 |
326 |
310 |
376 |
429 |
322 |
362 |
453 |
419 |
|
2 |
403 |
591 |
658 |
440 |
506 |
482 |
445 |
445 |
444 |
|
3 |
954 |
916 |
984 |
768 |
866 |
874 |
546 |
614 |
539 |
|
4 |
990 |
918 |
980 |
700 |
680 |
644 |
301 |
579 |
529 |
|
5 |
305 |
443 |
508 |
347 |
489 |
440 |
192 |
298 |
283 |
|
6 |
350 |
494 |
663 |
392 |
504 |
468 |
217 |
402 |
391 |
|
7 |
336 |
383 |
391 |
406 |
599 |
670 |
321 |
464 |
542 |
|
8 |
504 |
640 |
625 |
447 |
744 |
668 |
354 |
637 |
625 |
|
9 |
1063 |
1484 |
1577 |
915 |
2460 |
2486 |
716 |
1020 |
1079 |
|
10 |
521 |
712 |
632 |
487 |
961 |
935 |
517 |
1228 |
1087 |
|
  |
Wanjiazhai |
Sanmenxia |
Xiaolangdi |
||||||
|
Maximum Water Level (m)a |
980.01 |
319.42 |
273.5 |
||||||
|
Minimum Water Level (m) |
921.38 |
283.46 |
205.01 |
||||||
a Water level data selected between April 30th, 2006 and December 31st, 2022.”
- In section 5.4, it analyzed computation results "Scheme A138 intersects the actual operation data, increasing the average output by 13.34% and reducing the average change in ecological flow by 11.81%. Scheme B66 intersects with the actual operation data, increasing the average output by 10.40% and reducing the average change in ecological flow by 29.05%. Scheme C33 intersects with the actual operation, increasing the average output by 16.36% and reducing the average change degree in ecological flow by 43.99%". How to evaluate the reduction of average change in ecological flow and the increase of average output? Which scheme is the optimal scheme for this multi-objective optimal problem?
Authors’ reply: The two goals set in this article are average output and average change in ecological flow. A higher average output represents better economic benefits, while a lower average change in ecological flow represents better ecology. The optimized results of this article are an increase in average output and a decrease in average change in ecological flow, both of which are better than the actual scheduling process. Secondly, this article selects three typical years, namely the year of high flow/sediment, median flow/sediment, and low flow/sediment. The optimization results of these three years are optimized separately. Therefore, this article focuses on these three typical years, and the optimization results are A138, B66, and C33, respectively.
- In the first paragraph of the introduction, it mentioned "reservoir construction can disrupt the natural equilibrium of the natural river flow structure and riverbed morphology". However, readers cannot find relationship between natural disruption and ecologic flow.
Authors’ reply: Thank you for the suggestions provided by the reviewers. We have reorganized this section and made modifications around the theme of the article. The specific modifications are as follows:
“Reservoirs mainly undertake a variety of important tasks such as flood control, power generation and water supply, but there is a synergistic competition relationship between different objectives. How to scientifically and reasonably allocate water re-sources and achieve the maximum total benefits of reservoirs is the goal pursued by managers[1-6]reservoir construction can disrupt the natural equilibrium of the natural river flow structure and riverbed morphology. In sediment-rich rivers, reservoirs are mainly constructed to intercept large amounts of sediment upstream to ensure the normal function of downstream rivers; however, this leads to the siltation of reservoir capacity and seriously affects the service life of reservoirs[7].”
- In figure 2. there are two points at the left of the picture. How to deal with these two points? Besides, figure 3. & figure 4. also exist some points away from point cluster. How to explain this phenomenon?
Authors’ reply: This article adopts the SA-NSGA-II algorithm, which is closely related to the selection of the initial population. Due to the uneven generation of the initial population, there are missing segments on the Pareto front. However, overall, the distribution of this point is on the Pareto front, so it is believed that although the points are discontinuous, knowledge can be trusted.
- English language is not fine.
Authors’ reply: Before submitting the manuscript, we have contacted the relevant polishing agency to modify the language of the article. If it still does not meet the requirements of your journal, we can choose to continue polishing on the website recommended by MDPI.
Reviewer 2 Report
The article is very interesting. It needs clarification and improvement of certain elements.
1. Too little literature cited in the text.
2. What range of hydrological years were taken into account in determining maximum minimum flow and median flow.
3. Differences between the schemes should be further explained, e.g. table 8,9. It is difficult to say what the differences between the different schemes A1 and A138 are. From the text it seems to depend on the values of the indicators or indices. Do these indices and the ecological index have any variable in the form of an equation, a range of data?
4. How were the data related to the sediment taken into account?
5. Why the method is geared towards the least possible loss energy and not the greatest possible positive ecological effect.
6. Lack discussion of results.
7. Summary written too generally. Some details from the results should be added and reference should be made to the years analysed.
Author Response
- Too little literature cited in the text.
Authors’ reply: We have reorganized the references of the article and made relevant supplements. The current reference is about twice that of the original article. Please refer to the revised manuscript for details.
- What range of hydrological years were taken into account in determining maximum minimum flow and median flow.
Authors’ reply: We deeply apologize for not clarifying the source and scope of the data before. We have supplemented the source of the data, the specific form of the data, and the selected time frame as follows:
“The optimal operation of the cascade reservoir group in the middle reaches of the Yellow River in three kinds of typical years (high flow/sediment year, median water/sediment year, and low flow/sediment year) were selected (The data selection time range is from April 30th, 2006 to December 31st, 2022). Among them, November 2008 to October 2009 was selected for the high flow/sediment year, November 2010 to November 2011 was selected for the median water/sediment year, and November 2016 to October 2017 was selected for the low flow/sediment year. The relevant data of the Wanjiazhai, Sanmenxia and Xiaolangdi Reservoirs were obtained from the hydrological yearbook of the Yellow River basin, as shown in Table 1, in which the suitable ecological flow of the downstream river during each period (month) of each reservoir was calculated by the monthly frequency computation method.
Table 1. The mean monthly discharge, upper limit of water level, and lower limit of water level in Wanjiazhai, Sanmenxia, and Xiaolangdi reservoirs
|
Typical year |
Years with high flow/sediment (2008.11-2009.10) |
Years with median flow/sediment (2010.11-2011.10) |
Years with low flow/sediment (2016.11-2017.10) |
||||||
|
mean monthly discharge(m3/s) |
Wanjiazhai |
Sanmenxia |
Xiaolangdi |
Wanjiazhai |
Sanmenxia |
Xiaolangdi |
Wanjiazhai |
Sanmenxia |
Xiaolangdi |
|
11 |
515 |
734 |
758 |
451 |
558 |
504 |
289 |
458 |
442 |
|
12 |
249 |
313 |
336 |
360 |
546 |
444 |
392 |
528 |
494 |
|
1 |
290 |
326 |
310 |
376 |
429 |
322 |
362 |
453 |
419 |
|
2 |
403 |
591 |
658 |
440 |
506 |
482 |
445 |
445 |
444 |
|
3 |
954 |
916 |
984 |
768 |
866 |
874 |
546 |
614 |
539 |
|
4 |
990 |
918 |
980 |
700 |
680 |
644 |
301 |
579 |
529 |
|
5 |
305 |
443 |
508 |
347 |
489 |
440 |
192 |
298 |
283 |
|
6 |
350 |
494 |
663 |
392 |
504 |
468 |
217 |
402 |
391 |
|
7 |
336 |
383 |
391 |
406 |
599 |
670 |
321 |
464 |
542 |
|
8 |
504 |
640 |
625 |
447 |
744 |
668 |
354 |
637 |
625 |
|
9 |
1063 |
1484 |
1577 |
915 |
2460 |
2486 |
716 |
1020 |
1079 |
|
10 |
521 |
712 |
632 |
487 |
961 |
935 |
517 |
1228 |
1087 |
|
  |
Wanjiazhai |
Sanmenxia |
Xiaolangdi |
||||||
|
Maximum Water Level (m)a |
980.01 |
319.42 |
273.5 |
||||||
|
Minimum Water Level (m) |
921.38 |
283.46 |
205.01 |
||||||
a Water level data selected between April 30th, 2006 and December 31st, 2022.”
- Differences between the schemes should be further explained, e.g. table 8,9. It is difficult to say what the differences between the different schemes A1 and A138 are. From the text it seems to depend on the values of the indicators or indices. Do these indices and the ecological index have any variable in the form of an equation, a range of data?
Authors’ reply: Although there is a competitive relationship between power generation efficiency and ecological efficiency, due to the large amount of water in the year of high water and sand, even if the scheme with the highest power generation efficiency is chosen, the average ecological change value is relatively small, and the impact on ecology is also small. Moreover, the ecological efficiency is still greater than that in the year of median flow/sediment, and low flow/sediment. Moreover, the conclusion obtained from the constructed evaluation model is still that A138 is the best, which means that in rainy years, multiple occurrence points can be selected to achieve the optimal comprehensive benefits.
- How were the data related to the sediment taken into account?
Authors’ reply: This article did not consider sediment, mainly because the Yellow River Basin conducts water and sediment regulation every year. This activity is mainly aimed at achieving the reduction of sedimentation in the studied reservoirs and the problem of erosion in the lower reaches of the Yellow River. Our research aims to study the joint operation problem of different water and sediment years, only discussing the two goals of power generation efficiency and ecological efficiency. In the next step, we aim to analyze the coupling of sediment discharge factors, Build a multi-objective scheduling model for power generation, ecology, and sediment reduction.
- Why the method is geared towards the least possible loss energy and not the greatest possible positive ecological effect.
Authors’ reply: At present, the main tasks of reservoirs in the middle reaches of the Yellow River are flood and sediment transport, socio-economic and ecological environment. However, due to the shortage of water resources in the Yellow River Basin, priority is given to meeting the requirements of flood and sediment transport, followed by socio-economic and ultimately ecological environment under limited water resources. Therefore, this article focuses on optimizing the three reservoir joint debugging scheme, and the priority of power generation efficiency should be higher than the ecological environment. From this perspective, when constructing the evaluation index system, the proportion of power generation efficiency should be higher than the ecological efficiency.
- Lack discussion of results.
Authors’ reply: The author has added relevant discussions on the results, with specific comments on lines 403-421 and 566-587
“From the above results, it can be shown that the average output and ecological flow changes of reservoirs in years with high flow/sediment, median flow/sediment, and low flow/sediment generally exhibit a competitive relationship. When reservoirs pursue power generation benefits, they will sacrifice some ecological benefits. Moreover, this competitive relationship is closely related to the water volume, such as significant dif-ferences in overall power generation efficiency in years with high flow/sediment, me-dian flow/sediment, and low flow/sediment, ranging from 2.8759~3.2014×105kW, 2.6493~2.8267×105kW, 2.5540~2.5518×105kW respectively. The ecological benefits have the same pattern, with the average change range of the three typical annual ecological flows being 0.0746~0.1389, 0.1908~0.2587, and 0.2691~0.2653, respectively. The overall ecological benefits are still closely related to annual runoff, and there is more water available for regulation in high flow/sediment year, so the degree of regulation is also the highest, followed by median flow/sediment year and the worst in low flow/sediment years. Overall, annual runoff is the most important factor affecting power generation and ecological benefits. The more water there is, the greater the de-gree of regulation and the greatest benefits generated. However, due to the competitive relationship between power generation efficiency and ecological benefits, there is a trend of eliminating each other. Therefore, how to scientifically select the most suitable scheduling plan from the optimal solution set is worth in-depth research.
The proposed evaluation decision-making method based on grey target theory and cumulative prospect theory-fuzzy evaluation method, which introduces grey target theory and cumulative prospect theory, uniformly quantifies the established scheme evaluation index system, and can effectively deal with uncertain issues such as cognitive limitations and bounded rationality of decision-makers. The fuzzy evaluation method is used to optimize the reservoir operation plan, reducing the subjectivity of the decision-making process as much as possible. This method can comprehensively consider the impact of different weighting methods and decision-maker bounded rationality on the evaluation results, and the recommended scheme can better reflect the multi-objective scheduling purpose, resulting in more reasonable results.
In addition to flood control and sedimentation reduction during the flood season, the reservoirs in the Yellow River Basin, along with other reservoirs in the upper reaches of the Yellow River, also undertake various scheduling tasks such as ecology, water supply and irrigation. Selecting a relatively simple objective function for comprehensive power generation benefits and ecological benefits can be applicable to the operation of cascade reservoirs during non flood seasons; However, for the Yellow River, which is severely short of water resources, with severe soil erosion, and frequent downstream water disasters, a multi reservoir joint scheduling approach should be adopted to establish a multi-objective scheduling model for reservoir groups that comprehensively considers flood control, power generation, sediment reduction, ecology, and water supply irrigation, and explore its efficient solution methods. Relevant research work will be carried out in the future.”
- Summary written too generally. Some details from the results should be added and reference should be made to the years analysed.
Authors’ reply: Thank you for the reviewer's suggestion. We have made modifications. Please refer to line 589-611:
(1) SA-NSGA-II, a two-objective optimal operation model of the Wanjiazhai, Sanmenxia, and Xiaolangdi cascade reservoir group in the middle reaches of the Yellow River, is solved for three typical years, with multiple Pareto optimal operation sets and formulating three reservoir operation scheme sets, including 138, 111 and 91 operation schemes, respectively. The operation scheme set fully utilizes the water resources of the middle reaches of the Yellow River by optimizing the ecological benefits of the downstream river channel, while ensuring the output of the reservoir.
(2) This paper illustrates the construction of a comprehensive benefit evaluation index system for reservoir operation schemes, based on the three first-level indices of power generation, ecology and flood control. By integrating each reservoir’s data and combining the gray target theory and cumulative prospect theories, to obtain recommended schemes. Select schemes similar to A138, B66, and C33 in high flow/sediment, median flow/sediment, and low flow/sediment years, respectively.
(3) Through evaluation and optimization, six optimal schemes were obtained for each typical year. Compared with actual operation data. Scheme A138 increased the average output by 13.34% compared to actual operation, and decreased the average change in ecological flow by 11.81%; Scheme B66 intersects with actual scheduling, increasing the average output by 10.40% and reducing the average change in ecological flow by 29.05%; Scheme C33 intersects with actual scheduling, increasing the average output by 16.36% and reducing the average change in ecological flow by 43.99%, which can simultaneously balance power generation and ecological benefits. Provide a new decision-making approach and technical support for the optimal operation of reservoirs in the middle reaches of the Yellow River.”
Reviewer 3 Report
The aim of this work is to optimise a cascade of reservoirs, taking into account the ecological impact of reservoir operation as well as the economic benefit (i.e. hydropower generation). It is particularly important that the sediment transport is correctly considered in the multi-objective optimisation framework. The establishment of an evaluation index is one of the main contributions and key elements of this work. Therefore, I recommend this work for publication in WATER after addressing the following comments:
Comments:
1) What do the authors mean by "average performance" (line 125)?
2) Please support the sentence "Reservoirs mainly perform a variety of important tasks such as flood control, power generation and water supply, but reservoir construction can disrupt the natural balance of natural river flow structure and riverbed morphology" with the following references: "Detailed Simulation of Storage Hydropower Systems in Large Alpine Watersheds" and "Freshwater Ecosystems versus Hydropower Development: Environmental assessments and conservation measures in the transboundary Amur river basin".
3) Please support the sentence "Particle swarm optimisation" line 13 also with "A dual-layer MPI continuous large-scale hydrological model including Human Systems" and "J. Robinson, Y. Rahmat-Samii Particle swarm optimisation in electromagnetics".
4) Please emphasise either in the introduction or in the conclusion that the optimisation strategies are aimed at maximising energy production and not revenue, which is related to the energy market in a different context. See for example "Short-term hydropower optimisation driven by innovative time-adaptive econometric model".
Author Response
- What do the authors mean by "average performance" (line 125)?
Authors’ reply: This refers to the power generation efficiency, which is expressed using the average output. The article is not accurate and has been revised to 'average output'.
- Please support the sentence "Reservoirs mainly perform a variety of important tasks such as flood control, power generation and water supply, but reservoir construction can disrupt the natural balance of natural river flow structure and riverbed morphology" with the following references: "Detailed Simulation of Storage Hydropower Systems in Large Alpine Watersheds" and "Freshwater Ecosystems versus Hydropower Development: Environmental assessments and conservation measures in the transboundary Amur river basin".
Authors’ reply: The modifications have been made according to the suggestions.
Reservoirs mainly undertake a variety of important tasks such as flood control, power generation and water supply, but there is a synergistic competition relationship between different objectives. How to scientifically and reasonably allocate water re-sources and achieve the maximum total benefits of reservoirs is the goal pursued by managers[1-6].
[5]Simonov, E. A., Nikitina, O. I., & Egidarev, E. G. Freshwater ecosystems versus hydropower development: Environmental assessments and conservation measures in the transboundary Amur River Basin. Water, 2019, 11(8), 1570.
[6]Galletti, A., Avesani, D., Bellin, A., & Majone, B. Detailed simulation of storage hydropower systems in large Alpine watersheds. Journal of Hydrology, 2021, 603, 127125.
- Please support the sentence "Particle swarm optimisation" line 13 also with "A dual-layer MPI continuous large-scale hydrological model including Human Systems" and "J. Robinson, Y. Rahmat-Samii Particle swarm optimisation in electromagnetics".
Authors’ reply: The modifications have been made according to the suggestions.
Common solutions are mathematical programming methods and artificial intelligence optimization methods[23-24]. These include genetic algorithms [25-27], particle swarm optimization [28-29], artificial neural networks [30-31], ant colony optimization [32-33], simulated annealing [34-35], support vector machines [36], and fuzzy computation[37].
[28]Avesani, D., Galletti, A., Piccolroaz, S., Bellin, A., & Majone, B. A dual-layer MPI continuous large-scale hydrological model including Human Systems. Environmental Modelling & Software, 2021, 139, 105003.
- Please emphasise either in the introduction or in the conclusion that the optimisation strategies are aimed at maximising energy production and not revenue, which is related to the energy market in a different context. See for example "Short-term hydropower optimisation driven by innovative time-adaptive econometric model".
Authors’ reply: The modifications have been made according to the suggestions.
The purpose of reservoir optimization scheduling is to use optimization methods to formulate optimal scheduling methods for the inflow process and comprehensive utilization requirements of the reservoir, in order to achieve better benefits. The optimization strategy aims to maximize energy production rather than income, which is related to the energy market in different contexts[41]. At present, large-scale reservoir joint operation has become a key focus in the field of reservoir operation, with more and more constraints and objective functions. Therefore, the optimization of algorithms has become one of the bottlenecks that constrain joint operation.
[41] Avesani, D., Zanfei, A., Di Marco, N., Galletti, A., Ravazzolo, F., Righetti, M., & Majone, B. Short-term hydropower optimization driven by innovative time-adapting econometric model. Applied Energy, 2022, 310, 118510.
Round 2
Reviewer 2 Report
The authors responded to my comments in great detail. I think that the article in its present form is suitable for publication.